# Integrating AI with Advanced Hyperspectral Imaging for Enhanced Classification of Selected Gastrointestinal Diseases

**DOI:** 10.3390/bioengineering12080852

**Published:** 2025-08-08

**Authors:** Chu-Kuang Chou, Kun-Hua Lee, Riya Karmakar, Arvind Mukundan, Tsung-Hsien Chen, Ashok Kumar, Danat Gutema, Po-Chun Yang, Chien-Wei Huang, Hsiang-Chen Wang

**Affiliations:** 1Division of Gastroenterology and Hepatology, Department of Internal Medicine, Ditmanson Medical Foundation Chia-Yi Christian Hospital, Chia-Yi 60002, Taiwan; vacinu@gmail.com (C.-K.C.); 07742@cych.org.tw (P.-C.Y.); 2Obesity Center, Ditmanson Medical Foundation Chia-Yi Christian Hospital, Chia-Yi 60002, Taiwan; 3Department of Trauma, Changhua Christian Hospital, Changhua, No. 135, Nanxiao St., Changhua City 50006, Taiwan; 88847@cch.org.tw; 4Department of Mechanical Engineering, National Chung Cheng University, 168, University Rd., Min Hsiung, Chia Yi 62102, Taiwan; karmakarriya345@gmail.com (R.K.); d09420003@ccu.edu.tw (A.M.); 5Department of Biomedical Imaging, Chennai Institute of Technology, Sarathy Nagar, Chennai 600069, Tamil Nadu, India; 6Department of Internal Medicine, Ditmanson Medical Foundation Chia-Yi Christian Hospital, Chia-Yi 60002, Taiwan; cych13794@gmail.com; 7Department of Computer Science, Vel Tech Rangarajan Dr. Sagunthala R&D Institute of Science and Technology, No. 42, Avadi-Vel Tech Road Vel Nagar, Avadi, Chennai 600062, Tamil Nadu, India; ashokvijay872@gmail.com (A.K.); danatgutema@gmail.com (D.G.); 8Department of Gastroenterology, Kaohsiung Armed Forces General Hospital, 2, Zhongzheng 1st. Rd., Lingya District, Kaohsiung City 80284, Taiwan; 9Department of Nursing, Tajen University, 20, Weixin Rd., Yanpu Township, Pingtung 90741, Taiwan; 10Department of Medical Research, Dalin Tzu Chi Hospital, Buddhist Tzu Chi Medical Foundation, No. 2, Minsheng Road, Dalin, Chia-Yi 62247, Taiwan; 11Department of Technology Development, Hitspectra Intelligent Technology Co., Ltd., Kaohsiung 80661, Taiwan

**Keywords:** ulcerative colitis, polyps, esophagitis, gastrointestinal diseases, hyperspectral imaging, narrow-band imaging, white-light imaging

## Abstract

Ulcerative colitis, polyps, esophagitis, and other gastrointestinal (GI) diseases significantly impact health, making early detection crucial for reducing mortality rates and improving patient outcomes. Traditional white light imaging (WLI) is commonly used during endoscopy to identify abnormalities in the gastrointestinal tract. However, insufficient contrast often limits its effectiveness, making it challenging to distinguish between healthy and unhealthy tissues, particularly when identifying subtle mucosal and vascular abnormalities. These limitations have prompted the need for more advanced imaging techniques that enhance pathological visualization and facilitate early diagnosis. Therefore, this study investigates the integration of the Spectrum-Aided Vision Enhancer (SAVE) mechanism to improve WLI images and increase disease detection accuracy. This approach transforms standard WLI images into hyperspectral imaging (HSI) representations, creating narrow-band imaging (NBI-like) visuals with enhanced contrast and tissue differentiation, thereby improving the visualization of vascular and mucosal structures critical for diagnosing GI disorders. This transformation allows for a clearer representation of blood vessels and membrane formations, which is essential for determining the presence of GI diseases. The dataset for this study comprises WLI images alongside SAVE-enhanced images, including four categories: ulcerative colitis, polyps, esophagitis, and healthy GI tissue. These images are organized into training, validation, and test sets to develop a deep learning-based classification model. Utilizing principal component analysis (PCA) and multiple regression analysis for spectral standardization ensures that the improved images retain spectral characteristics that are vital for clinical applications. By merging deep learning techniques with advanced imaging enhancements, this study aims to create an artificial intelligence (AI)–driven diagnostic system capable of early and accurate detection of GI diseases. InceptionV3 attained an overall accuracy of 94% in both scenarios; SAVE produced a modest enhancement in the ulcerative colitis F1-score from 92% to 93%, while the F1-scores for other classes exceeded 96%. SAVE resulted in a 10% increase in YOLOv8x accuracy, reaching 89%, with ulcerative colitis F1 improving to 82% and polyp F1 rising to 76%. VGG16 enhanced accuracy from 85% to 91%, and the F1-score for polyps improved from 68% to 81%. These findings confirm that SAVE enhancement consistently improves disease classification across diverse architectures, offers a practical, hardware-independent approach to hyperspectral-quality images, and enhances the accuracy of gastrointestinal screening. Furthermore, this research seeks to provide a practical and effective solution for clinical applications, improving diagnostic accuracy and facilitating superior patient care.

## 1. Introduction

Gastrointestinal (GI) disease is an alteration of the course of the digestive tract from the mouth to the rectal opening. The complex and metabolically active ecosystem of the microbiome plays a crucial role in the alteration of organs like the esophagus, the stomach, the intestines (small and large), the liver, the pancreas, and the gallbladder [1]. Genetic predisposition to smoking initiation was associated with an increased risk of GI diseases, including 4 lower GI diseases (irritable bowel syndrome, diverticular disease, Crohn’s disease, and ulcerative colitis), 7 upper GI diseases (gastroesophageal reflux, esophageal cancer, gastric ulcer, duodenal ulcer, acute gastritis, chronic gastritis, and gastric cancer), 8 hepatobiliary and pancreatic diseases (non-alcoholic fatty liver disease, alcoholic liver disease, cirrhosis, liver cancer, cholecystitis, cholelithiasis, and acute and chronic pancreatitis), and acute appendicitis [2]. Even though there are many contributors to GI disease, in 2019, alcohol use was the largest contributor at the global level for both sexes [3]. With the study conducted in 33 countries with a minimum of 2000 individuals surveyed in each country, in both internet and household surveys, equal sex distribution (49.5% women, 50.5% men) in most countries with both surveying methods was achieved [4]. GI cancers accounted for nearly 26% of the worldwide cancer incidence in 2018, which was about 4.8 million new cases. These cancers also accounted for 35% of the total deaths related to cancer, which was approximately 3.4 million deaths around the globe. Most cases of new GI cancer cases (63%) and GI cancer deaths (65%) were from Asia, followed by Europe and North America, which only accounted for 26% of global cases and 23% of deaths [5]. In 2018, the most common GI cancer, with significant incidence rates globally, was colorectal cancer [6]. As epidemiologic data indicates, the majority of the cases are sporadic and lack heredity or any family background; as a result, microbial, lifestyle, nutritional, and environmental factors play important roles in the increasing number of victims of colorectal cancer [7].

The main disorders of the colon and rectum are ulcerative colitis and polyps, as shown in Figure 1. Figure 1 shows four different types of esophageal images: normal tissue, ulcer, polyp, and esophagitis. These images are captured using white light imaging (WLI) technology and are used to diagnose GI diseases. Below is a detailed description of each image: The Normal Squamous Mucosa image shows the normal tissue of the esophagus, which is characterized by smooth and uniform folds. This image is often used to contrast diseased tissue and help doctors identify abnormalities. The Ulcer image shows a distinct area of mucosal loss with clear boundaries. This lesion is often associated with inflammation or infection, and early detection is critical to prevent further deterioration. The Polypoid Lesion image shows a polypoid lesion protruding from the lumen. Polyps can be benign or malignant, and their size and shape are important indicators of their potential risk. The Esophagitis image shows redness and edema of the mucosa, which is often related to acid reflux or other irritants. This lesion may cause difficulty swallowing and chest pain. Figure 1 shows the application of WLI technology in the diagnosis of GI diseases, but its contrast is limited, and it is difficult to distinguish subtle mucosal and vascular abnormalities. This also highlights the need for the introduction of more advanced imaging technologies such as SAVE. Ulcerative colitis (UC) is a chronic inflammatory disease of the colon and rectum, which can cause common symptoms like bloody diarrhea, abdominal pain, and weight loss. These diseases can be complicated by being persistently and acutely inflamed and ulcerated. Patients with UC are at an increased risk of developing colorectal cancer over time, making early diagnosis crucial [8]. The main characteristic that makes UC different from other inflammatory bowel diseases (IBD), such as Crohn’s disease, includes continuous inflammation and being limited to the mucosal layer [9]. Polyps are abnormal tissue growths that develop on the lining of the colon and rectum. Most of the polyps are benign; however, certain types, like adenomatous polyps, have the behavior to convert into colorectal cancer if not detected and treated at the early stage. Polyps are different in size, shape, and histological characteristics, with larger polyps having a higher risk of malignancy [10]. To reduce this malignancy and the risk of colorectal cancer development, the early detection and removal of polyps through colonoscopy or AI-assisted image analysis is crucial [11]. The other GI disease is Esophagitis, which is mainly the inflammation of the lining of the esophagus. It is commonly caused by gastroesophageal reflux disease (GERD), allergies, infections, or prolonged exposure to irritants such as alcohol and certain medications [12]. Difficulties in swallowing, chest pain, and heartburn are the main symptoms of esophagitis. The precancerous condition increases the risk of esophageal cancer, such as esophageal strictures or Barrett’s esophagus, and the chronic result of esophagitis [13]. To visualize the entire GI tract and detect conditions like ulcerative colitis, polyps, and esophagitis, the non-invasive method called wireless capsule endoscopy is used [14]. However, traditional WLI often has the drawback of struggling with contrast limitations, making it difficult to detect subtle mucosal abnormalities [15].

Over the past decade, artificial intelligence (AI) and machine learning (ML) have played an important role in advancing medical diagnostics, particularly in GI diseases. For instance, making use of it with enhanced colonoscopy in the detection of colorectal cancer has raised polyp detection rates, overcoming limitations for better patient outcomes [16]. Similarly, AI applications in esophageal disease diagnostics have shown good results by accurately diagnosing eosinophilic esophagitis (EoE) through histopathological features, helping in its identification and treatment planning [17]. Moreover, deep learning-based models, such as convolutional neural networks (CNNs), have been used to predict colorectal cancer diagnosis using immunohistochemical staining image features, achieving high accuracy and offering the potential for better results [18]. A comprehensive review highlights the growing role of these methods in colorectal cancer diagnosis, emphasizing their power to outperform conventional histopathological techniques [19]. However, AI-based techniques have also been utilized for early identification of colorectal cancer for the given WLI images; the diagnostic capabilities remain limited, indicating that WLI imaging alone is insufficient for accurate detection of abnormalities. Being less sensitive to point out the mucosal and vascular changes that are indicative of early cancer was the main drawback of WLI images, which are captured with a wide range of spectrum [20]. Hence, integrating HSI to capture images with a specific range of spectrum combined with an AI model was a much more promising way. This paper makes four important contributions to the field of endoscopic image-based GI disease screening:Converting hyperspectral-inspired imaging to software: In this study the Spectrum Aided Vision Enhancer (SAVE), a completely software-based post-processing pipeline that changes standard white light endoscopic frames into NBI, HSI images was developed. This gives better contrast between the mucosa and blood vessels without needing any special equipment.Spectral Standardization Based on Clinical Needs: Created a two-step calibration system that combines principal component analysis (PCA) with multiple regression. This allows SAVE products to reproduce important spectral signatures that can be used to evaluate pathology.The Deep Learning Multi-Class GI Classification: Built and improved a CNN classifier based on InceptionV3 and EfficientNet backbones that uses SAVE-enhanced images to accurately identify ulcerative colitis, polyps, esophagitis, and healthy mucosa.Strict Measurement and a Proven Increase in Performance: Statistically significant binary performance improvements in overall accuracy, sensitivity, and specificity on a chosen dataset of paired WLI and SAVE images, even when compared to models trained only on WLI that had not been enhanced.

The rest of this paper is structured as follows. Section 2 describes our proposed Spectrum Aided Vision Enhancer (SAVE) pipeline along with the description of the dataset and the design of the deep learning classifier built on InceptionV3 and EfficientNet backbones. Section 3 presents our quantitative evaluation metrics and comparative results demonstrating the performance gains afforded by SAVE. Section 4 outlines the limitations and the future scope of the study. Finally, Section 5 concludes the paper.

## 2. Related Works

HSI is an emerging medical imaging technique that captures and processes information across the electromagnetic spectrum, providing detailed spectral and spatial data about biological tissues. HSI operates by acquiring images at numerous narrow wavelength bands, producing a three-dimensional dataset—known as a hypercube—with two spatial dimensions (X and Y) and one spectral dimension (λ) PMC [21]. The HSI device captures images with hundreds of spectral bands, and this rich spectral information enhances the contrast differentiation between normal and abnormal tissues, thus actually facilitating better detection and diagnosis. The primary HSI acquisition methods can be classified into spectral scanning, spatial scanning, and snapshot techniques. Spectral scanning captures images sequentially at different wavelengths. In contrast, spatial scanning acquires spatial information line by line across the scene, while snapshot techniques capture the whole spectral and spatial information in one exposure for real-time imaging [22,23]. For instance, the study used HSI with deep learning (DL) to accurately identify the geographical origins of *Lycium barbarum* L. (*L. barbarum*, *Goji berry*), 97.79% accuracy was achieved with improvements of 4.41% and 5.14% [24]. Another work shows that contact-free HSI has potential utility as a novel tool for free real-time monitoring of human pancreatic grafts, which could improve the outcome of pancreas transplantation [25]. However advantageous these methods may be, they mostly require costly, specialized equipment like advanced cameras and accurate calibration systems, which becomes even more difficult to afford and complex to run in a clinical environment [26]. For this issue, NBI was developed as an alternative that is far more suitable for practical applications. NBI utilizes specific wavelength bands to enhance the visualization of mucosal and vascular patterns without the need for complex instrumentation [27]. It makes use of a specific narrow spectrum of light blue (415 nm) to green (540 nm), which has a power penetrating superficial layers of mucosa [28]. This increases superficial tissue contrast, assisting in the detection of lesions through light filtration within narrow bands. Recent developments make it possible to produce endoscopic NBI-like images from regular WLI by way of hyperspectral data, which improves NBI-like image quality and diagnostic accuracy during endoscopic procedures [29,30]. A study that focuses on describing sinonasal mucosa under white light and NBI of 103 patients (82 evaluated) with 29 sinonasal pathologies and 55 controls (33 evaluated), 80.6% vs. 90.6% specificity, and 53% vs. 80.3% accuracy was calculated for both WLI and NBI consecutively [31]. Hence, in this study, NBI has been coupled with HSI to selectively augment certain spectral bands, creating a contrast enhancement for the identification of colorectal diseases and esophagitis in earlier stages. Using the SAVE model, this work will also train various ML algorithms applied to the dataset so as to create an effective system for the detection and prevention of precancerous and inflammatory conditions. The WLI images from the dataset have been SAVE-transformed to allow future training and analysis. NBI and hardware-driven HSI endoscopy can improve contrast, but they require specialized equipment and come with high costs and integration problems. Researchers have looked into methods that are only based on software, but they haven’t fully studied them in a rigorous way to predict spectrum standardization or a multi-type disease of GI cancer. In this study, using a software-only SAVE to fill in this gap is suggested. It uses standard WLI to create HSI-inspired images, adds a PCA and regression-based calibration to maintain clinical spectral characteristics, and works with deep learning classifiers to give a correct diagnosis of GI disease in four categories.

## 3. Materials and Methods

### 3.1. Mathematical Problem Statement

Let the dataset be denoted as: **𝒟** = { (x_i_, y_i_) }_i=1_^n^, where each input image x_i_ ∈ ℝ^H×W×3^ is a white-light endoscopy (WLI) image, and the label y_i_ ∈ {1, 2, 3, 4} represents one of four GI conditions: ulcerative colitis, polyps, esophagitis, or healthy tissue. The goal is to learn a spectral enhancement transformation function, T_SAVE: ℝ^H×W×3^ → ℝ^H×W×c^, where C > 3 denotes the number of spectral channels after enhancement. This transformation augments the standard RGB WLI image into a spectrally enriched representation with improved contrast and enhanced diagnostic features. Subsequently, a classifier, f_θ: ℝ^H×W×c^ → {1, 2, 3, 4}, parameterized by θ, is trained on the enhanced representations. The objective is to minimize the categorical cross-entropy loss: L(θ) = −(1/N) ∑_i=1_^n^ ∑_K__=__1_^4^
1 [y_i_ = k] log p_K_^(i)^, where p_K_^(i)^ is the predicted probability for class k on sample i, and 1 [·] is the indicator function. To ensure spectral fidelity in the transformed images, a spectral standardization step is applied, involving principal component analysis (PCA) followed by multivariate regression. This step preserves diagnostically relevant spectral features that are not present in conventional RGB space. Thus, the joint optimization problem consists of learning the enhancement function SAVE to generate clinically informative spectral representations and training the classifier to maximize diagnostic accuracy across all GI conditions.

### 3.2. Dataset

For an objective, head-to-head evaluation, we utilized solely the publicly available KVASIR benchmark dataset, comprising 6000 WLI endoscopic images, as the assessment set. We applied our SAVE enhancement pipeline to each provided WLI frame without implementing any specific exclusions or filters, resulting in 6000 pairs of WLI and SAVE images that align precisely with the original benchmark. All modeling, hyperparameter tuning, and final testing were conducted on these predefined subsets to ensure complete reproducibility of our results and facilitate comparison with any prior or subsequent methods applied to the same public data. The dataset used in this work includes WLI and SAVE images across four classes: ulcerative colitis, polyps, esophagitis, and normal GI images. The images were acquired from KVASIR, which is captured via wireless capsule endoscopy (WCE), ensuring high-quality and well-annotated samples. The original dataset consists of a total of 6000 images and was divided into train (800 images per class), Val (500 images per class), and test (200 images per class). After applying different preprocessing tasks including the removal of noise the dataset contains a training set of 300 images per class (with a total of 1200 images for WLI), a validation set of 100 images per class (with a total of 400 images for WLI), and a test set of 100 images per class (with the total 400 images for WLI) with the ratio of 80%, 10%, and 10%. The final selected WLI images were converted to SAVE images, leading to a result with a total of 1200 training images, 400 validation images, and 400 test images, as shown in Figure 2.

### 3.3. SAVE

This research thoroughly focused on a spectrum-helped vision enhancer (SAVE) mechanism, which is a helpful technique that uses a standard endoscope to convert WLI to HSI and then SAVE images. The main goal of the SAVE technique is to improve endoscopy’s diagnostic abilities with very detailed information that customary endoscopy cannot provide. The study focuses on matching the reflectance spectra from a spectrometer by calibrating RGB (Red, Green, Blue) images caught by an endoscope. Each calibration process is completed by the Macbeth Color Checker (X-Rite Classic, Grand Rapids, MI, USA), which has twenty-four color patches that fully represent natural colors. This tool is very important to create a special correlation that connects each image of WLI with the spectrometer output. Images caught within the standard WLI (sRGB) color space are always converted initially to CIF 1931 XYZ color space, employed universally throughout color science. In the XYZ color space, X, Y, and Z always stand for the colors red, green, and blue, respectively. For accurate color calibration (XYZcorrect), corrections are applied to the endoscope images for non-linear response, dark current, and incorrect color separation, including color distortion. This helps achieve the desired result. Equations (1) and (2) represent the function used to calculate the XYZcorrect after error correction is utilized.(1)$$\left[C\right]=\left[{XYZ}_{spectrum}\right]\times pinv\left(\left[V\right]\right)\;$$(2)$$\left[{XYZ}_{correct}\right]=\left[C\right]\times \left[V\right]\;$$

In the calibration process, it is important to guarantee a close match between each of the images of the SAVE and all of the images of the NBI, done by using the X-Rite Color Checker and caught by Olympus’s endoscope (Tokyo, Japan). Principal component analysis (PCA) helped in the determination of all exceptionally important components found in the reflectance spectra information; this subsequently reduced each dimension of the data and considerably improved computational efficiency. The PCA results indicated that six principal components accounted for practically all (99.64%) of the variance in the data. This quantity was certainly sufficient for a color calibration that was both effective and accurate. A minimal color difference was seen. The calibration recorded a mean root mean square error (RMSE) of 0.19 between the calibrated color (XYZcorrect) along the spectrometer output (XYZspectrum). This calibration process is quite important in converting all WLI images to HSI images and then transforming all of them into NBI images, which stimulates an improved level of tissue visualization, similar to narrow-band imaging. Multiple regression analysis was done to further assist in changing images. The transformation matrix equation in 3 helped to thoroughly define the correlation among all of the variables. This important transformation matrix adjusts the image spectra to obtain SAVE from WLI images.(3)$$\left[M\right]=\left[Score\right]\times pinv([V_{color}])$$

Equation (4) supports the accuracy of the system by showing negligible color differences when compared to actual measurements:(4)$$\left[{SS}_{pectrum}{]}_{380\sim 780}=[EV][M][V_{color}\;\right]$$

After converting all of the RGB images to HSI images and developing all of the reflectance data, the conventional types of images should have all of the NBI images simulated, and precise band selection to detect all types of cancer should be done. For this study, the Olympus endoscope was used with a 24-color checker for color calibration purposes, since it has a reference for NBI catch mode that helps in comparing the developed algorithm. It is decidedly important to guarantee that all SAVE images from the HSI conversion method possess a special similarity. These images should mirror all NBI images taken by the Olympus endoscope. Next, the SAVE from the HSI conversion algorithm was carefully compared to the actual NBI images caught by the Olympus endoscope. The CIEDE 2000 color variation was found for each of the 24 colors, and the mean color variation was calculated to be 2.79, which indicated very little difference. Color variation between the SAVE and the NBI is mainly due to the light spectrum, the reflection spectrum, and the color-matching function. Since the light spectrum used throughout the SAVE images from HSI conversion, and the Olympus endoscope NBI, were dissimilar, a large difference between those two images occurred during each initial CIEDE 2000 color variation process between every SAVE image and each Olympus endoscope NBI image. The study introduced an additional calibration step using the Cauchy-Lorentz distribution, which significantly reduced the color variation between NBI images. This step also greatly minimized the color variation between images of SAVE, especially in the 450–540 nm range. This distribution is shown in Equation (5), and it is useful for a complete analysis of the system’s spectral response.(5)$$f\left(x;x_{0},\gamma \right)=\frac{1}{\pi \gamma [1+(\frac{x-x_{0}}{\gamma }{)}^{2}]}=\frac{1}{\pi }\left[\frac{\gamma }{(x-x_{0}{)}^{2}+\gamma^{2}}\right]\;$$

For optimization of the light spectrum, the dual annealing optimization algorithm is used; it is derived from all generalized simulated annealing algorithms (every simplified classical simulated annealing (CSA) plus every fast simulated annealing (FSA)). The mean standard CIEDE 2000 color variation was accurately recorded as 5.36. This is a completely negligible value. Since the Olympus endoscope showed a color predominantly of brown rather than green and blue (hemoglobin wavelength, 415 and 540 nm), making NBI images appear considerably more realistic through image post-processing becomes exceptionally helpful. To evaluate the SAVE mechanism’s effectiveness, as well as its calibration process, three different image quality metrics were used; these include entropy, Peak Signal-to-Noise Ratio (PSNR), and Structural Similarity Index Metric (SSIM). SSIM shows how similar the SAVE images are. It also shows similarities between the NBI images of the Olympus endoscope. SSIM distinctly indicates a large resemblance of 94.27% between the SAVE images and the NBI images, so the SAVE mechanism is more reliable for carefully simulating NBI imaging. Each image texture’s randomness or complexity was measured by entropy. Olympus endoscope NBI had a difference of precisely 0.37% with SAVE on average. The PSNR values, which measure reconstructed image quality in image shrinking and denoising applications, confirmed the simulated images’ high fidelity; it was recorded as 27.8819 dB for the Olympus NBI. Figure 3 shows the complete process of converting WLI to SAVE images. This process is achieved through the SAVE mechanism, which aims to improve the contrast and diagnostic accuracy of the images.

### 3.4. ML Algorithm

In this study, the InceptionV3 and EfficientNet backbones were chosen for three reasons:Certified Medical Imaging: Both architectures have shown cutting-edge performance on a variety of medical tests that classify features and provide a stable extraction of subtle vascular and mucosal patterns.Capture of features at multiple scales: InceptionV3’s mixed-scale inception modules are great at showing both fine-grained texture and coarser structural clues, which are important for understanding SAVE-enhanced images.Quickness: EfficientNet’s compound scaling method strikes a good balance between model complexity and inference speed, which is important because the model will need to be used in real time in clinics in the future.

All models were trained utilizing the Adam optimizer with an initial learning rate of 3 × 10^−4^, β_1_ = 0.9, β_2_ = 0.999, and without weight decay. We conducted training for 300 epochs, without employing an early-stopping criterion while monitoring validation loss. A batch size of 32 was employed, and input images were resized to 640 by 640 pixels. Learning rate scheduling was executed using ReduceLROnPlateau. The training set underwent data augmentation through random horizontal and vertical flips, rotations of ±15°, and brightness and contrast adjustments of ±10%. All remaining parameters were maintained at their default library settings.

#### 3.4.1. Inceptionv3

Inceptionv3 is one of the deep convolutional network (CNN) image classifiers that incorporates several architectural improvements to enhance efficiency and accuracy [32]. A number of smaller convolutions, like two 3 × 3 convolutions, fully replace larger convolutions, like 5 × 5. Factorized convolutions provide a method for reducing all computing expenses, without changing any receptive field [33]. To further optimize performance, asymmetric convolutions, like dividing n × n into 1 × n and n × 1, reduce the total number of parameters. Another key feature is auxiliary classifiers; these are small networks put in at layers in the middle, and they act as regularizers to make the vanishing gradient problem smaller and to help convergence get better when training. For training, the network uses categorical cross-entropy loss, which is given by:(6)$$L=\;-\sum_{i=1}^{N}y_{i}log\left(x_{i}\right)\;$$
where *y_i_* stands for real class label, and *x_i_* is the predicted probability for each class *I* [34].

To stop the model from being too confident in its predictions, label smoothing is used a lot. Label smoothing adjusts the ground truth labels by distributing a small probability mass to incorrect classes.(7)$$y_{i}^{\prime }=y_{i}\left(1-\epsilon \right)+\frac{\epsilon }{N}\;$$
where *ϵ* (epsilon) helps prevent overfitting by making the model less certain about its prediction. These optimizations make inception-v3 a highly effective model for complex image classification tasks [35].

#### 3.4.2. VGG16

VGG16 is a deep CNN architecture developed by Simonyan and Zisserman. It has a design that is both simple and effective, and it has sixteen weight layers (thirteen convolutional layers and three fully connected layers) [36]. Each layer uses small 3 × 3 convolutions stacked consecutively so it can catch detailed spatial features. The number of parameters is also kept controllable. Within the hierarchical network structure, pooling layers gradually reduce spatial dimensions, and fully connected layers perform classification at the end [37]. VGG16 is easy to implement since it uses only convolutional layers, unlike newer setups. However, it is computationally expensive due to its large number of parameters and the lack of residual connections or beginning modules [38]. 

VGG16 is trained using the categorical cross-entropy loss, which is given by:(8)$$L=\;-\sum_{i=1}^{N}y_{i}\log\left(x_{i}\right)$$

For each class *i*, *y_i_* represents the true label, while xi represents the expected likelihood. Batch normalization is widely used in modified VGG16 versions to stabilize training and reduce internal covariate shifts, hence significantly boosting generalization. Due to its characteristic of learning numerous hierarchical feature representations fast, VGG 16 has been widely used as a feature extractor in transfer learning. Despite the fact that a variety of architectures like ResNet and EfficientNet outperform VGG16 in terms of efficiency, it remains a widely used benchmark model in deep learning research [39].

#### 3.4.3. YoloV8x

YOLOv8x, also known as You Only Look Once version 8 extra-large, is a state-of-the-art deep learning model that has been additionally improved for image classification along with object detection. YOLOv8 is one component of the YOLO series with different variants (YOLOv8n, YOLOv8s, YOLOv8m, YOLOv8l, and YOLOv8x). A few architectural enhancements, like a better CSPDarknet backbone, dynamic anchor-free detection, plus more feature fusion techniques, are introduced in it, and a measurable level of accuracy and efficiency results [40,41]. Because YOLOv8x generalizes better than previous models, it is better suited to the real-time classification of medical images. YOLOv8x backbones use Cross-Stage Partial Networks (CSPNet) to improve gradient flow and lessen computational redundancy, without losing accuracy [42]. The model also uses a path aggregation network (PANet) to better find detailed features in medical images through improved multi-scale feature fusion. YOLOv8x uses a single-stage detection framework, unlike typical CNN-based classifiers, so it is very efficient for real-time inference but still has good classification performance. YOLOv8x uses binary cross-entropy loss for classification and CIoU loss for localization during training, defined as:(9)$$L_{CIoU}=1-IoU+\;\frac{\rho^{2}(b,\;b^{gt})}{C^{2}}+\;\alpha v\;$$
where *IoU* represents the intersection-over-union between predicted and ground-truth bounding boxes, *ρ*^2^(*b*,*b^gt^*) gauges Euclidean distance between the center points, *C* stands for the diagonal length of the smallest enclosing box, and *v* denotes a shape consistency term [43,44]. This loss function improves the precision and robustness of YOLOv8x in detecting and classifying medical anomalies. Furthermore, YOLOv8x incorporates efficient activation functions such as SiLU (Swish Linear Unit) instead of ReLU, improving gradient propagation and convergence speed during training [45]. These optimizations allow YOLOv8x to outperform previous YOLO versions, making it a powerful tool for medical image classification tasks.

## 4. Results

Due to the evaluation being performed on a singular predefined test set without repeated resampling, present only point estimates; confidence intervals and *p*-values are not relevant in this context. Future endeavors will encompass repeated cross-validation and bootstrapping to facilitate formal statistical significance testing. Evaluation metrics are mathematical, quantitative, and objective measures for grading the accuracy of performance of the efficacy of statistical or MI algorithms. These provide very important information with respect to understanding how the model is performing, help in the comparison of different models, or return that could be the same model with the same algorithm configuration. Accuracy measures the percentage of samples in the data set that were correctly categorized by the model as positive, thus measuring the capacity of the model to avoid false positives, as shown in Equation (10).(10)$$Accuracy=\frac{TP+TN}{TP+TN+FP+FN}\;$$

Recall computes the ratio of correct positive cases relative to all the cases that are positive actually; it deals with the capacity of the model to diagnose all instances of a particular class as shown in Equation (11).(11)$$\begin{array}{rr} \;Precision\; & =\frac{tp}{tp+fp}\\ \;Recall\; & =\frac{tp}{tp+fn} \end{array}$$

The F1 Score, which is formed by the precision score and the recall score, and summing them. The F-1 Score, therefore, delivers a more accurate account of the performance of the model in the balanced mode, either of false positives or false negatives, as shown in Equation (12).(12)$$F=2\cdot \frac{\;precision\times \;\;recall\;}{\;precision+recall\;}$$

Recall ranges from 0 to 1, and the calculation of the mean of average precisions was done. The mAP formula depends upon the following sub-metrics: Confusion matrix, Intersection over Union IoU, Miss rate, Hit rate, as shown in Equation (13). Four characteristics are required to develop the confusion matrix, which are True Positives, True Negatives, False Negatives, and False Negatives. mAP50 is interpreted as the Mean average precision, calculated at the intersection over union of 0.50. It has something to do with how well the model performs based on the subset of detections that every algorithm should ideally be able to handle easily. mAP50–95 is the mean average precision at the intersection of the union of different levels of IoU thresholds from 0.50 to 0.95. Therefore, it gives an overview of the performance of the model with respect to object findability in an image.(13)$$mAP=\frac{1}{N}\sum_{i=1}^{N}\;{AP}_{i}\;$$

All models, including those utilizing WLI and those enhanced by SAVE, demonstrated significant convergence in our experiments, well before the 250-epoch threshold: validation loss stabilized at a plateau approximately at epoch 280. All the backbones exhibited a training accuracy to validation accuracy surpassing 97%. This indicates the absence of significant overfitting.

For the case of the InceptionV3 model, even though the overall accuracy of both WLI and SAVE images is the same at 94%, accuracy alone does not provide a complete evaluation of the model’s ability, as shown in Table 1. The class-wise performance shows that the SAVE model has a significantly higher recall value (99%) for the UC class than that of WLI (92%), indicating that the model captures more actual positive cases with SAVE. Moreover, being 1% higher in the F1 score of the UC class in SAVE suggests a more balanced trade-off between precision and recall. However, the precision for Polyps is much higher in SAVE (99%) compared to WLI (93%), but its recall is lower (76% vs. 82%), which implies that while SAVE is more confident in its positive predictions, it misses more true cases. The F1 scores for the Esophagitis class remain similar for both imaging techniques, highlighting the model’s robustness in detecting this condition. These variations suggest that hyperparameter tuning, such as optimizing learning rates, batch sizes, dropout rates, and incorporating additional data augmentation techniques, could further enhance the model’s ability to generalize across different classes, ultimately improving its performance with SAVE images.

The YOLOv8x model yields better accuracy on SAVE images (89%) compared to WLI images (79%), showing that the model performs better on SAVE images. This means that some characteristics are more clearly shown in the SAVE imaging system, giving the YOLOv8x model a better chance to make correct predictions. For Esophagitis, the model recalls more true positive instances in the WLI system than in the SAVE system, with a recall of 94% for WLI compared to 88% for SAVE. By contrast, F1-scoring remained stable for both (97% for WLI and 96% for SAVE), meaning Esophagitis can be diagnosed reliably with either imaging method. With SAVE scoring higher recall and F1 score than WLI in the detection of UC, the SAVE approach captures more true classes and predisposes the model to balanced predictions. This indicates that the model’s recognition of UC instances with SAVE images is enhanced because SAVE provides better contrast or sharper feature representation. When analysing the Polyps class, the performance is mixed. Precision is higher in SAVE (99%) compared to WLI (93%), meaning that the model is more confident in its positive predictions with SAVE images. Even though, the recall is lower in SAVE (76%) compared to WLI (82%), implying that while YOLOv8x is more confident in predicting Polyps from SAVE images, the harmonic value of both precision and recall which is the f1 score is much higher in SAVE images, indicating the model is identifying actual positive cases well while minimizing false negative and false positive.

For the VGG16 model, the overall accuracy is 85% for WLI and 91% for SAVE images, with significant variations in class-wise performance. For the overall result of precision, recall, and F1 score, the SAVE model outperforms WLI, indicating that SAVE can capture more positive cases and reduce the false positive classes. Having lower precision in WLI (78%) than in SAVE (87%) for the polyp’s class indicates that the model makes fewer false positive predictions in WLI. WLI has a much lower recall (60%) than SAVE (75%), meaning that VGG16 misses more actual polyp cases in WLI. These results show that SAVE allows the model to detect more true polyp cases at the cost of slightly lower precision, as shown in Figure 4. For the Esophagitis class, both imaging techniques perform equally well, with F1-scores of 98% in WLI and 100% in SAVE, showing that SAVE is slightly effective in identifying Esophagitis. From these analyses, it is clear that SAVE images generally provide better recall, particularly for UC and Polyps, meaning that VGG16 detects more true cases with this imaging type.

A Friedman Test was conducted to assess the repeated effect of the performance disparity among models and imaging strategies, analyzing the F1-scores across all classes for each model and imaging combination. The findings yielded a statistically significant result regarding the differences between the comparative groups (X0 = 13.13, *p* = 0.0222). Indicating that none of the model-imaging combinations exhibited comparable performance. Subsequently, we conducted a Nemenyi post hoc test to identify the differences present in specific pairings. The test indicated that YOLOv8x-WLI consistently underperformed compared to InceptionV3-WLI and InceptionV3-SAVE, with *p*-values of 0.087, approaching the threshold of statistical significance. Comparisons between YOLOv8x-SAVE and YOLOv8x-WLI (*p* = 0.469) indicated a numerical enhancement in performance, yet failed to reach the significance threshold of 0.05. The remaining two-sided tests produced results indicating *p* > 0.25, signifying no significant difference (see Appendix A for the outcomes of the Nemenyi test, which are illustrated in a heat map, highlighting the distinctiveness of the YOLOv8x-WLI relative to other configurations). The finding is statistically significant, indicating that the observed improvements are attributable to the SAVE enhancement, particularly in models with subpar baseline results in standard white light imaging. The findings confirm that SAVE is effective in improving the accuracy of various model architectures’ diagnoses.

## 5. Discussion

There is a lot of research that shows how standard CNN architectures can be used to analyze GI images. However, this is the first study to show a SAVE pipeline that works only with software. It changes the traditional WLI in endoscopic images into NBI, which is inspired by HSI without needing any special hardware. This study uses two-stage PCA spectral standardization based on PCA and regression to keep only the spectral features that are important for clinical purposes. It can be linked to deep learning models to make it possible to classify GI diseases into four strong classes. Currently, no one has yet put together end-to-end software HSI transformation, strict spectral calibration, and multi-class classification into a single framework. So, there are no published studies that compare paired WLI/HSI datasets to our results. Instead, SAVE is compared to a baseline model that was trained on raw WLI images from the same dataset. Based on the data in Table 1, through the horizontal comparative analysis of models such as InceptionV3, VGG16, and YOLOv8x, we can explore which model is most suitable for detecting specific diseases (such as ulcerative colitis or polyps). First, in the detection of ulcerative colitis, InceptionV3 showed the best performance under the SAVE technology, with a recall of 99%, indicating that the model can effectively detect most cases of ulcerative colitis. In addition, the F1-score of InceptionV3 is 93%, which performs best in the balance of accuracy and sensitivity and is suitable for the diagnosis of early ulcerative colitis. In contrast, the recall of VGG16 under the SAVE technology is 89%, which is an improvement, but still lower than that of InceptionV3; YOLOv8x’s performance is relatively weak, with a recall of only 78% and an F1-score of only 82%, which is not as stable as other models. Therefore, InceptionV3 is the first choice for ulcerative colitis detection. Secondly, in the detection of polyps, InceptionV3 performed well under WLI technology, with a Recall of 82% and a Precision of 93%, indicating that the model can maintain high sensitivity and accuracy when detecting polyps. However, under SAVE technology, the Recall of InceptionV3 dropped to 76%. Although the Precision increased to 99%, the overall performance (F1-score of 86%) was slightly lower than that of WLI technology. In contrast, VGG16’s performance under SAVE technology improved, with a Recall of 75%, a Precision of 87%, and an F1-score of 81%, which was better than WLI technology overall. YOLOv8x’s Recall under SAVE technology was 83%, which was significantly improved from 57% under WLI technology, but its Precision was 71%, its F1-score was 76%, and its overall performance was still inferior to InceptionV3 and VGG16. Therefore, in polyp detection, InceptionV3 performs best under WLI technology, while VGG16 improves significantly under SAVE technology. Overall, InceptionV3 is the most suitable model for detecting ulcerative colitis, especially under SAVE technology, which can provide high sensitivity and accuracy. In polyp detection, InceptionV3 performs more balanced under WLI technology, but SAVE technology has a more significant effect on the performance improvement of VGG16. Future research can consider further exploring the performance of other models (such as ResNet50 or EfficientNet) to optimize the detection ability of specific diseases. The *p*-value is used to test whether the performance difference between models is statistically significant. When the *p*-value is less than 0.05, it indicates that the performance difference between models is significant. According to the data in Table 1, if the *p*-values of the models under different technologies (such as WLI and SAVE) are all lower than 0.05, it can be considered that the application of technology has a significant impact on the performance of the model. In addition, the confidence interval (CI) provides the range of variation of the performance index. If the confidence intervals of the two models do not overlap, it further supports the significance of the performance difference. For example, if the Recall of InceptionV3 under SAVE technology is 99% (95% CI: 97–100%), and the Recall of VGG16 is 89% (95% CI: 85–92%), and the *p*-value is <0.05, it can be considered that the detection ability of InceptionV3 is significantly better than that of VGG16. At the same time, the size of the confidence interval also reflects the stability of the data. A narrower interval indicates that the model’s performance is more stable.

The results of this study thoroughly indicate that the SAVE model, when applied to colorectal and esophageal disease detection, performs standard imaging techniques, especially in considerably improving early identification. This improvement in appearance might provide measurable advantages to patients since finding possible or real cancer earlier allows doctors to respond faster, which lowers the odds of the illness worsening. A main problem with this study, though, is the information, since it is from just one source. Expanding the dataset through the incorporation of images from many hospitals as well as varied geographical regions, in addition to several demographic groups—containing differing racial groups along with different age groups—would greatly improve the model’s generalizability and also guarantee it performs sufficiently well across populations. This may explain different genetic and ecological changes, which are important in how colorectal and esophageal diseases show up. Even though the dataset mostly looks at a few kinds of colorectal cancer and esophagitis, future work needs to check out more conditions. For example, in colorectal cancer, other types such as serrated polyps and hyperplastic polyps could be checked to see how well the SAVE model functions in telling apart different kinds of lesions that might become cancer. Likewise, a sense of exactly how well the model performs at detecting additional kinds of problems would come from using it on esophageal diseases, along with Barrett’s esophagus, high- and low-grade dysplasia, as well as adenocarcinoma. Barrett’s esophagus comes before esophageal adenocarcinoma and is often linked to lasting acid reflux. Because of this, finding it early is key to treatment to help prevent it. The inclusion of these conditions in future research would help evaluate the SAVE model’s performance across a broader diagnostic spectrum. To assess the effectiveness of the SAVE model in comparison to conventional WLI imaging, three different ML models were trained and tested. These models—YOLOv8x, InceptionV3, and VGG16—were chosen for their well-documented performance in medical image analysis. Among them, inceptionv3 exhibited the highest accuracy and reliability, making it the most effective model for this task. However, the results suggest that further optimization of the other models could lead to improved performance. Future studies should consider training additional ML architectures, including transformer-based models like Vision Transformers (ViTs) and more advanced convolutional networks, to explore their potential advantages in medical imaging. Additionally, fine-tuning hyperparameters and optimizing the preprocessing techniques used in the SAVE transformation process could further enhance detection accuracy.

## 6. Conclusions

This study demonstrates the effectiveness of the SAVE mechanism in enhancing the diagnostic accuracy of GI diseases by converting traditional WLI into spectral-assisted visual enhancement (SAVE) images. Through advanced spectral analysis and MI algorithms, SAVE significantly improves the visualization of mucosal and vascular structures, addressing the limitations of conventional WLI. The integration of deep learning models, such as InceptionV3 and EfficientNet, further enhances the system’s ability to classify and detect various pathologies, including ulcers, polyps, and esophagitis. The experimental results highlight the potential of SAVE as a reliable tool for early disease detection, offering a cost-effective and efficient alternative to narrow-band imaging (NBI). Future research should focus on expanding the dataset to include more diverse cases, optimizing the model architecture for real-time clinical applications, and exploring the applicability of SAVE in other medical imaging domains. By bridging the gap between traditional imaging techniques and advanced computational methods, this study paves the way for more accurate and accessible diagnostic solutions in gastroenterology, ultimately improving patient outcomes and reducing healthcare costs. In conclusion, SAVE is an efficient and cost-effective software enhancement for endoscopic imaging. While InceptionV3 achieved an accuracy of 94%, SAVE yielded measurable advantages in specific pathologies, notably enhancing F1 tags for ulcerative colitis by 1%. Significant improvements were observed in YOLOv8x (accuracy +10%, ulcerative colitis F1 + 27%) and VGG16 (accuracy + 6%, polyp F1 + 13%), demonstrating the capability of SAVE to highlight diagnostically relevant features. SAVE enables the earlier and more accurate identification of GI disease using narrow-band contrast, without requiring specialized equipment or additional radiation exposure.

## Figures and Tables

**Figure 1 bioengineering-12-00852-f001:**
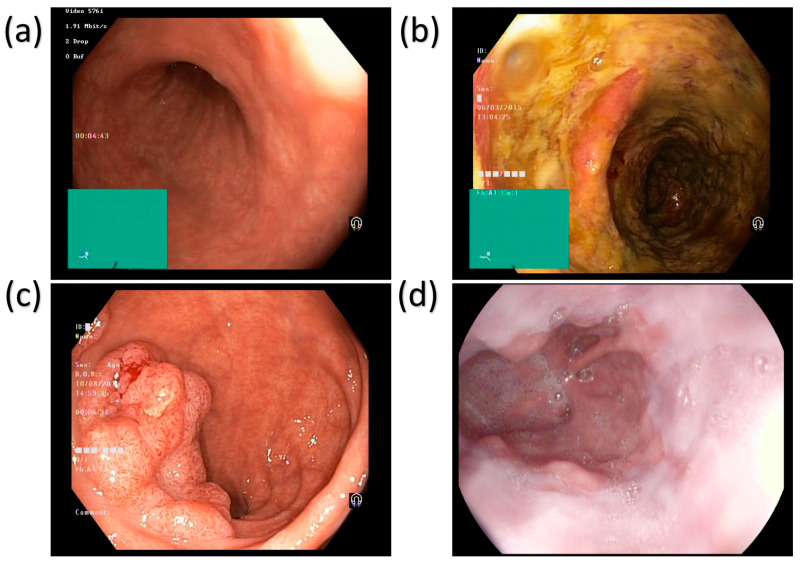
WLI images of the esophagus demonstrating common findings. (**a**) Normal squamous mucosa with smooth, uniform folds; (**b**) Ulcer with a well-demarcated mucosal defect; (**c**) Polypoid lesion protruding into the lumen, and (**d**) Esophagitis characterized by erythema and mucosal edema.

**Figure 2 bioengineering-12-00852-f002:**
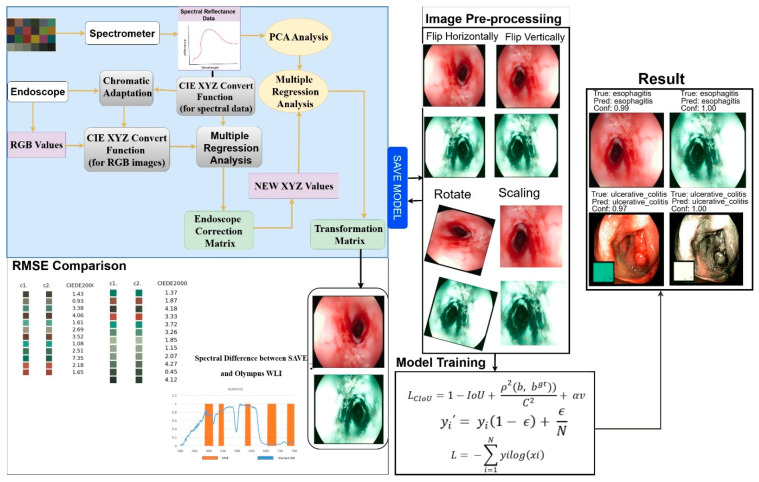
The overall flowchart of the project.

**Figure 3 bioengineering-12-00852-f003:**
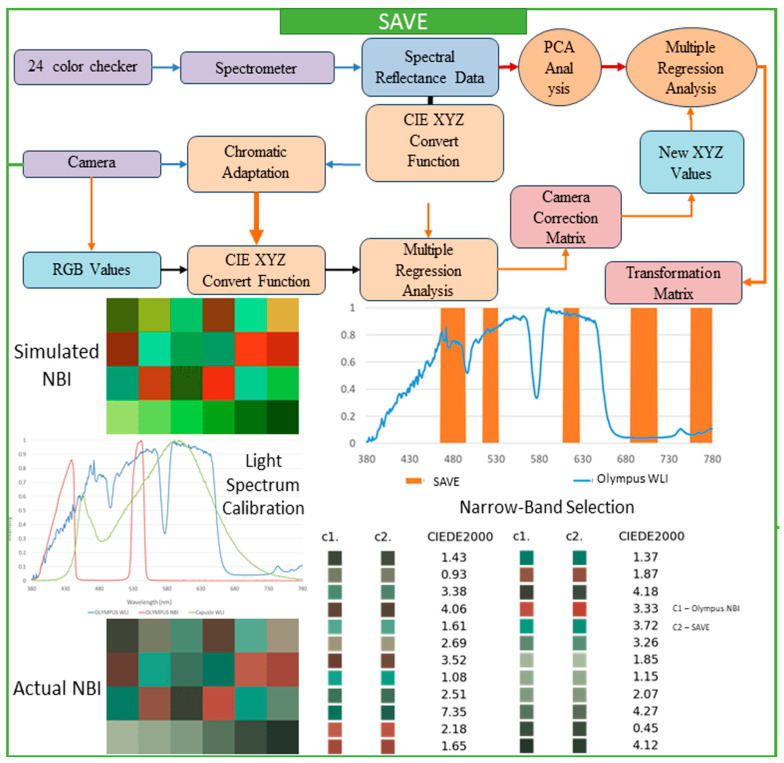
The SAVE Image—Transformation Pipeline.

**Figure 4 bioengineering-12-00852-f004:**
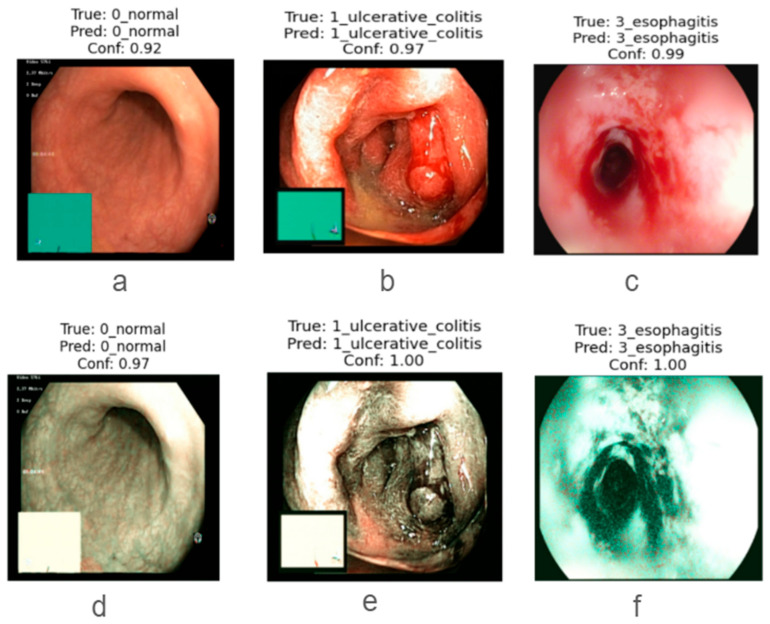
Representative endoscopic frames showing the effect of SAVE enhancement across three GI categories. Top row: original white-light images of (**a**) healthy mucosa, (**b**) ulcerative colitis, and (**c**) esophagitis. Bottom row: corresponding SAVE-enhanced outputs for (**d**) healthy mucosa, (**e**) ulcerative colitis, and (**f**) esophagitis. All images were processed by the InceptionV3-based classifier, illustrating how SAVE improves contrast and delineation of mucosal and vascular features critical for accurate disease detection.

**Table 1 bioengineering-12-00852-t001:** The overall results of three different ML algorithms.

ML Models	Imaging Techniques	Classes	Precision	Recall	F1-Score	Accuracy
InceptionV3	WLI	Normal	93%	100%	96%	94%
		Ulcerative Colitis	91%	92%	92%	
		Polyps	93%	82%	87%	
		Esophagitis	97%	100%	99%	
	SAVE	Normal	93%	99%	96%	94%
		Ulcerative Colitis	87%	99%	93%	
		Polyps	99%	76%	86%	
		Esophagitis	98%	100%	99%	
YOLOV8x	WLI	Normal	100%	88%	94%	79%
		Ulcerative Colitis	39%	91%	55%	
		Polyps	78%	57%	66%	
		Esophagitis	100%	94%	97%	
	SAVE	Normal	100%	96%	98%	89%
		Ulcerative Colitis	86%	78%	82%	
		Polyps	71%	83%	76%	
		Esophagitis	99%	99%	99%	
VGG16	WLI	Normal	81%	100%	89%	85%
		Ulcerative Colitis	85%	82%	84%	
		Polyps	78%	60%	68%	
		Esophagitis	96%	99%	98%	
	SAVE	Normal	88%	100%	94%	91%
		Ulcerative Colitis	88%	89%	89%	
		Polyps	87%	75%	81%	
		Esophagitis	100%	100%	100%	

## Data Availability

The data presented in this study are available in this article upon considerable request to the corresponding author (H.-C.W.).

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
