# Peer review of "Integrating AI with Advanced Hyperspectral Imaging for Enhanced Classification of Selected Gastrointestinal Diseases"

_bioengineering, 2025, doi:10.3390/bioengineering12080852_

Round 1

Reviewer 1 Report

Comments and Suggestions for Authors

The paper is devoted to the investigation of integrating the Spectrum-Assisted Vision Enhancer mechanism with advanced hyperspectral imaging to enhance the classification of gastrointestinal diseases selected. The study suggests a method for transforming standard white light imaging images into hyperspectral representations to improve the visualization of critical vascular and mucous structures for diagnosing gastrointestinal disorders. It corresponds to the aim and scope of the "Bioengineering" journal.

There are the following comments.

  1. In the introduction, it is necessary to clearly and structure indicate the new scientific results (contributions) obtained in the manuscript.

  2. At the end of the introduction, authors need to add the structure of the manuscript.

  3. The introduction seems to be overloaded with information on some details from relevant fields. I suggest the authors add a separate "Related Works" section. Additionally, it is essential to explicitly state which research gap the authors are addressing in this paper.

  4. The problem statement must be explicitly given. Section 2 immediately begins with datasets. It is necessary to structure the material clearly in this section: problem - method of solution - features of solution, and then proceed to the results.

  5. The scientific novelty of this research or, rather, the proposed methodology is not clear. All algorithms and neural networks used are standard and well-known. There are newer YOLO architectures. The authors need to clearly explain the technical novelty of the solution and explain the motivation for choosing specific neural networks.

  6. Section 3 (Results) begins with a table. At the beginning of this section, it is necessary to include some introductory text. The accuracy metrics from Table 1 have been determined, but it would be helpful to provide explicit formulas for them. Additionally, it is recommended to highlight the best values in the table in bold.

  7. The manuscript does not compare the accuracy obtained by other researchers who use other methods or neural networks for their research on the datasets discussed in this paper.

Reviewer 2 Report

Comments and Suggestions for Authors

The papaer by Chou et al describes application of AI for the classification of images of different gastrointestinal diseases. Theme  of the study is important and interesting, however, the authors failed to demonstrate details of the study.

  1. The authors divided data into training, validation and test, but there is no results for these groups.
  2. The authos performed single division of the data, but what will happen if other portion of data will apper in training?
  3. May the authors somehow prove that the proposed technique utilizes useful feautes and not only noises?
  4. The abstract and conclusions are too general and missing exact obtained values.
  5. May the data from obe sample appear both in tarining and test sets?
  6. The authors must present learning curves.

The paper may not be published in the present form.

Reviewer 3 Report

Comments and Suggestions for Authors

The manuscript is largely sound in methodology and presentation, but would benefit from minor revisions related to language clarity, figure presentation, and dataset detail.

  1. Clarify the final dataset size more clearly: the flow from 6000 raw images to preprocessed subsets is not entirely intuitive on first read.
  2. Consider including a visual schematic or brief pseudocode of the SAVE image transformation pipeline for better understanding.

  3. Hyperparameter details for the ML models (e.g., learning rate, epochs, batch size) should be briefly summarized.

  4. Avoid overly emphatic or subjective terms such as “absolutely outperforms,” “decidedly more complete,” or “considerably benefit.” A more neutral academic tone is preferred.

  5. Consider professional editing to improve sentence flow and reduce redundancy in some paragraphs.

  6. Figure 3 could be improved in resolution and accompanied by clear labeling of the image classes and model used.

  7. Statistical significance (p-values and CIs) are referenced conceptually but not explicitly shown. Including exact values would improve transparency. 

Round 2

Reviewer 1 Report

Comments and Suggestions for Authors

A number of comments remain:

  1. In section 1, in spite of the authors' comments, there was no mathematical problem statement. A general discussion was not enough.
  2. The conclusion about the statistical significance of the results, which is usually associated with hypotheses, p-values and other factors, should be clarified by the authors. There are specific statistical tests, such as Friedman's, Nemeny's and Dunn-Sidak's, etc., for such cases.
  3. Related works should be in a separate section. It is not a part of the materials and methods.

All other responses are acceptable.

Reviewer 2 Report

Comments and Suggestions for Authors

The authors addressed some issues, but still there are two major questions without answers:

  1. There is no comparison of accuracies for training, validation and test;
  2. May the authors somehow prove that the proposed approach utilizes useful features of the images? May the proposed approach utilize random features of the analyzed images?

Moreover, equations 10-13 belongs to methods.

The paper may not be published in the present form.

Reviewer 3 Report

Comments and Suggestions for Authors

Addressed all comments. 

Round 3

Reviewer 1 Report

Comments and Suggestions for Authors

All responses are clear. In the mathematical formulation, it is necessary to refine expressions so that upper and lower indices are correctly written. In addition, I recommend including in the manuscript figures that are provided as supplementary materials. Firstly, they can be inserted directly into the text. Secondly, authors can create an Appendix with them.

Reviewer 2 Report

Comments and Suggestions for Authors

The authors addressed arised issues and the paper may be published.
